# Multidisciplinary Approach in the Treatment of Descending Necrotizing Mediastinitis: Twenty-Year Single-Center Experience

**DOI:** 10.3390/antibiotics11050664

**Published:** 2022-05-16

**Authors:** Angela De Palma, Mirko Girolamo Cantatore, Francesco Di Gennaro, Francesca Signore, Teodora Panza, Debora Brascia, Giulia De Iaco, Doroty Sampietro, Rosatea Quercia, Marcella Genualdo, Ondina Pizzuto, Giuseppe Garofalo, Fabio Signorile, Davide Fiore Bavaro, Gaetano Brindicci, Nicolò De Gennaro, Annalisa Saracino, Nicola Antonio Adolfo Quaranta, Gianfranco Favia, Giuseppe Marulli

**Affiliations:** 1Unit of Thoracic Surgery, Department of Emergency and Organ Transplantation, University of Bari “Aldo Moro”, Piazza Giulio Cesare 11, 70124 Bari, Italy; m.cantatore19@studenti.uniba.it (M.G.C.); f.signore1@studenti.uniba.it (F.S.); dorapanza91@gmail.com (T.P.); deborabrascia@gmail.com (D.B.); g.deiaco24@gmail.com (G.D.I.); dorotysampietro@gmail.com (D.S.); r.quercia.78@gmail.com (R.Q.); margenualdo@gmail.com (M.G.); ondi@hotmail.it (O.P.); giugaro75@icloud.com (G.G.); giuseppe.marulli@uniba.it (G.M.); 2Clinic of Infectious Diseases, Department of Biomedical Science and Human Oncology, University of Bari “Aldo Moro”, Piazza Giulio Cesare 11, 70124 Bari, Italy; francesco.digennaro1@uniba.it (F.D.G.); fabiosignorile@yahoo.it (F.S.); davidebavaro@gmail.com (D.F.B.); gaetanobrindicci@gmail.com (G.B.); nico84degennaro@gmail.com (N.D.G.); annalisa.saracino@uniba.it (A.S.); 3Otolaringology Unit, Department of Basic Medical Science, Neuroscience and Sensory Organs, University of Bari “Aldo Moro”, Piazza Giulio Cesare 11, 70124 Bari, Italy; nicolaantonioadolfo.quaranta@uniba.it; 4Complex Unit of Odontostomatology, Interdisciplinary Department of Medicine, University of Bari “Aldo Moro”, Piazza Giulio Cesare 11, 70124 Bari, Italy; gianfranco.favia@uniba.it

**Keywords:** descending necrotizing mediastinitis, early diagnosis, surgical treatment, thoracotomy, cervicotomy, antimicrobial therapy, antibiotics

## Abstract

Descending necrotizing mediastinitis (DNM) is an acute, rare, severe condition with high mortality, but the optimal management protocol is still controversial. We retrospectively analyzed the results of multidisciplinary management in patients treated for DNM at our center over the last twenty years. Fifteen male patients, mean age 49.07 ± 14.92 years, were treated: 9 with cervico-pharyngeal etiopathogenesis, 3 peri-tonsillar/tonsillar, 2 odontogenic, 1 post-surgical; 6 with DNM type I, 6 with type IIA, and 3 with type IIB (Endo’s classification). Mean time between diagnosis and treatment was 2.24 ± 1.61 days. In all cases, mediastinum drainage via thoracotomy was performed after neck drainage via cervicotomy, associated with tooth treatment in two; one required re-operation; tracheostomy was necessary in 9, temporary intensive care unit stay in 4; 6 developed complications, without post-operative mortality. Main isolated germs were Staphylococci and Candida; 7 had polymicrobial infection. The most used antibiotics were meropenem, metronidazole, teicoplanin, third-generation cephalosporins and clyndamicin; anti-fungal drugs were fluconazole, caspofungin and anidulafungin. On multivariate analysis, presence of cardiovascular disease was statistically significantly associated with longer chest tube duration and hospital stay. DNM requires early diagnosis and treatment to reduce mortality and morbidity. The most effective treatment should provide a multidisciplinary approach, combining cervicotomy and thoracotomy to drain all infectious collections with administration and monitoring of the proper antimicrobial therapy.

## 1. Introduction

Descending necrotizing mediastinitis (DNM) is an acute and rare serious condition with a high mortality rate (ranging from 11% to 40%) that represents, in most cases, a complication of oro-pharyngeal or cervico-fascial infections, caused by both anaerobic and aerobic bacteria [1,2,3,4,5,6]. The diagnostic criteria of DNM, established by Estrera and colleagues, consider different aspects: clinical signs of severe infection, specific radiological signs, intra-operative and/or autoptic evidence of mediastinal infection and a strict etiopathogenetic relationship between oro-pharyngeal/cervical infection and mediastinitis [6]. The refinement of diagnostic techniques, such as neck and chest computerized tomography (CT), has significantly improved the prognosis of these patients, allowing an early diagnosis, staging (by Endo’s classification [7,8]) and treatment, which in most cases requires a surgical approach (combined cervicotomy and thoracotomy) with the aim of obtaining a complete cervical, mediastinal and pleural toilette, associated with antibiotic therapy with coverage for aerobic and anaerobic bacteria; nevertheless, due to the rarity and the complexity of DNM, the optimal management protocol is still controversial and only some recommendations but no guidelines have been developed to date [1,5,9]. In this retrospective study, we analyzed the results obtained with multidisciplinary treatment of patients affected by DNM at our center over the last twenty years; a secondary aim was to identify some clinical characteristics and risk factors that may interfere with the course of the disease and need further investigation in a multicenter setting.

## 2. Results

Fifteen patients affected by DNM were treated at our center from 1 January 2002 to 31 December 2020; all were males with a mean age of 49.07 + 14.92 years (range 12–70 years). They developed DNM with various etiopathogenesis: 9 had a cervico-pharyngeal origin of infection, 3 peri-tonsillar/tonsillar, 2 odontogenic and one originated as a post-operative complication (hemiglossectomy). 

The degree of diffusion of DNM according to Endo’s classification, defined using CT scans and intra-operative findings (Figure 1), was distinguished as follows: type I in 6 patients, type IIA in 6 and type IIB in 3. 

At the diagnosis of DNM, septic shock was already present in only two patients. 

In all cases, apart from clinical signs of severe infection, diagnostic certainty was achieved with contrast enhanced neck and chest CT, which allowed precise definition of the location and extent of the infectious process and therefore the best therapeutic strategy in a mean time of 2.24 ± 1.61 days (range 8 h–5 days). 

In all 15 cases it was necessary to combine a thoracotomy with a cervicotomy to obtain a complete cervical, mediastinal and pleural drainage (Figure 2); in particular, one patient underwent cervicotomy extended to pectoralis major and pectoralis minor muscles because there was a spread of the infective process also at this site; in two cases with odontogenic origin of the infection, treatment of the abscess of the diseased tooth was also carried out. 

Overall, 10 right lateral thoracotomies and 5 left lateral thoracotomies were performed. During thoracotomy, after opening the mediastinal pleura, mediastinal debridement and washing with intra-operative disinfectant solutions was carried out, as well as drainage and toilette of pleural collections and/or aspiration of any collections in other locations; material was collected for microbiological examination and large bore chest tubes were placed. 

Of the 15 cases analyzed, only one patient required re-operation: in particular, a right lateral re-thoracotomy was performed because of persistent signs of infection and post-operative CT scan evidence of a 7 cm loculated fluid collection with atypical arrangement, at the level of the right apical postero-mediastinal pleura. 

The mean duration of surgery was 130.63 ± 32.34 min (range 95–195 min). 

Tracheostomy, performed simultaneously in the thoracotomy/cervicotomy procedure, was necessary in 9/15 (60%) patients and an ICU stay in 4/15 (27%) patients. 

Post-operative complications occurred in 6/15 (40%) patients: according to the classification of surgical complications by Clavien-Dindo [10], there were 3 patients with grade II complications (hypoalbuminemia/dysprotidemia, associated with alteration of serum electrolytes in one, anemic state in one and anasarcatic state in another one), treated by pharmacological therapy and blood transfusion, 2 with grade IIIa complications (peri-tracheostomal bleeding in the thyroid region in one and persistent hydropneumothorax on the left and pleural effusion on the right side in another one), treated by interventions under local anesthesia, one with grade IIIa complications (partial thrombosis of left subclavian and brachiocephalic veins and respiratory failure), promptly treated and resolved with a short ICU stay.

The mean duration of chest (pleural/mediastinal) tubes was 18.91 ± 13.46 days (range 8–52 days).

The mean length of hospital stay was 29.33 ± 14.24 days (range 13–61 days) and the outcome of surgical treatment was favorable in all patients, with a post-operative mortality of 0%.

Regarding the type of germs isolated on the samples taken intra- or post-operatively (blood culture, bronchial lavage, pleural fluid), we observed, in 8/15 (53%) cases, bacteria members of the genus *Staphylococcus*, of which the most frequent were *S. epidermidis* (4 cases) and *S. aureus* (2 cases), but there was evidence also of *S. hominis* (one case) and *S. capitis* (one case); in 2/15 (13%) cases, bacteria belonging to the genus *Streptococcus*, in particular *S. anginosus* and *S. cristatus*; in 4/15 (27%) cases, bacteria belonging to other genera: *K. Pneumoniae*, *E. coli*, *S. maltophilia*, *Serratia marcescens* together with *P. aeruginosa*; in 3/15 (20%) cases, fungi were detected, in particular of the *Candida* genus: *Candida albicans* (2 cases) and *Candida parapsilosis* (one case). In 7/15 (47%) patients, there was a polymicrobial infection and three of them presented with both fungal and bacterial infection. Lastly, 4/15 (27%) patients were negative at microbiological examination. An antibiogram/antimicogram was performed in all positive cases. 

In all cases, antimicrobial post-operative therapy, on the basis of the antibiogram/antimicogram where available, was established after consultation with the infectious disease specialist, who also monitored the progress of treatment. The most commonly used antibiotics were meropenem, metronidazole, teicoplanin, third-generation cephalosporins and clyndamicin. For the microbiological positivity for fungi on intra-operative samples or blood cultures, anti-fungal drugs were also used, in particular fluconazole, caspofungin and anidulafungin. There were no complications related to the administration of antimicrobial treatments and no therapy had to be discontinued prematurely due to side effects from drugs. 

The mean body mass index (BMI) was 23.94 ± 2.67 (range 18.66–26.52), which is within the normal weight category. 

Considering pre-operative risk factors or co-morbidities, we observed smoking and/or COPD in 6/15 (40%) patients, overweight in 4/15 (27%), diabetes mellitus in 3/15 (20%), cardiopathy/vasculopathy in 3/15 (20%), chronic renal failure in 1/15 (7%), chronic liver disease in 1/15 (7%), other diseases in 4/15 (27%). On multivariate analysis, cardiovascular diseases resulted in statistically significant longer chest tube duration and hospital stay.

Data relating to the continuous variables considered in our study are summarized in Table 1, those relating to the nominal variables in Figure 3, those relating to the type of germs isolated and antimicrobial therapy in Table 2 and multivariate statistical analysis of variables in Table 3.

## 3. Discussion

In our study, concerning gender, we found DNM only in males, in agreement with some authors who described a higher incidence in males rather than females [9,11], but contrary to other authors who did not report any consistent differences between genders [5].

The mean age was about 50 years and this figure is consistent with the literature [5,9,12], even though cases in children of a few months and in the elderly have been described [5]; this wide age range is in agreement with our experience, since we treated a 12-year old child and three subjects over the age of 65 years.

Referring to etiopathogenesis, that is the site of origin and source of infection that evolved in DNM, we found an higher frequency for the cervico-pharyngeal (60%) site, followed by the tonsillar/peritonsillar (20%) and odontogenic (13%) sites; an iatrogenic source was found only in one case (7%), due to a hemiglossectomy that became complicated. These findings are only partially in agreement with the literature, because even if they are in line with the most frequent sites of origin (cervico-pharyngeal, tonsillar/peritonsillar and odontogenic) [1,5,12], most studies identified odontogenic infections as the main cause [13], in contrast to our study where they represent less than 1/6 of cases. A recent systematic review by Prado-Calleros et al. reported a steady reduction of the odontogenic causes (from 43% to 36%) and an increase in the cervico-pharyngeal (from 45% to 47%) ones [13]. This trend could be explained by the improved dental care techniques and the greater attention by both patients and doctors/odontostomatologists to oral hygiene and earlier symptoms of odontogenic diseases, which are therefore promptly treated, avoiding serious complications such as DNM.

The spread of infection was evaluated according to Endo’s classification of DNM [7,8]: type I (DNM localized in the upper mediastinum above the tracheal bifurcation); type IIA (DNM extends to the lower anterior mediastinum); type IIB (DNM extends to the anterior and lower posterior mediastinum). We found a prevalence of type II cases, overall representing 60%, and specifically, type IIA was found in 40% of patients, whereas type IIB in 20%; consequently type I was found in 40% of cases. However, contrary to what has been argued by some authors in other studies, where there was a clear prevalence or presence only of type II [3,4], we must point out that in our experience, type I was found in a high percentage, probably because of an early diagnosis with a timely treatment, that in fact was successful for all patients (0% mortality rate). 

Septic shock at the time of diagnosis of DNM was present only in 2 (13%) patients. This could be explained by the fact that one of them was the oldest of the patients (70-years old), whereas the other one had the most complex clinical history, since he had suffered from ischemic stroke and recurrent syncopal episodes, and was therefore a debilitated patient: therefore, these conditions could have favored the worsening of the infection and the consequent onset of the septic shock. However, this percentage seems to be lower than that reported in other studies, where death has been described after this dangerous complication [4,12]. Therefore, this is another factor that can explain the successful outcome of patients in our study; in fact, other authors reported that septic shock is an independent predictor of mortality [13] or otherwise associated with a higher morbidity [4].

The mean elapsed time between diagnosis of DNM and treatment was about two days, which is very short, similar to what other authors have described [12,13]. 

As for the surgical treatment, in all cases we performed thoracotomy after the cervicotomy carried out by otolaryngologysts (associated with diseased tooth treatment by odontostomatologists, in two cases with odontogenic origin), with the aim of draining and obtaining a toilette of pleural collections and a mediastinal debridement (after opening the mediastinal pleura); at the same time washing with intra-operative disinfectant solutions was performed. The prevalent thoracic approach was the right one (in 2/3 of cases), through lateral thoracotomy, which we preferred to the postero-lateral one as it is less demanding for the patient in terms of muscular section, ensuring a less painful and faster post-operative recovery for these patients, who were already immunocompromised by DNM. Standard thoracotomy in all its variants has high efficacy to treat DNM, according to many authors and in agreement with our experience [1,4,9].

In the literature, it is reported that in cases of type I DNM, drainage through cervicotomy is very effective [14]; however, if the infection condition persists, a more invasive approach is recommended [1,4]. This is in agreement with our experience, since in patients classified as Endo’s type I DNM, a first drainage through cervicotomy was carried out, but all of them presented with worsening in clinical and/or radiological conditions (post-operative neck-chest CT scan), hence the need for thoracotomy.

A re-operation was necessary in one case (7%) only, namely a right lateral re-thoracotomy, carried out two days after the first operation, due to persistent fever and post-operative CT scan evidence of a 7 cm loculated fluid collection at the level of the apical pleura. This case underlines the importance of CT scan in the follow-up, as the persistence of the infectious process was diagnosed early and treated, thus improving the prognosis of the patient. 

The mean duration of thoracic surgery was about two hours, and therefore relatively contained, which reduced the risk of intra and post-operative complications; despite this, complications occurred in the post-operative period in 40% of patients, although in all cases they were resolved with conservative treatment.

The post-operative complication rate in our study (40%) was similar to that of some authors [3], but relatively lower than others reporting percentages of about 65% or even more [4,12,13]. According to the classification of surgical complications by Clavien-Dindo [10], based on the type of therapy needed to correct the complication, the majority of our patients developed grade II complications, successfully treated by pharmacological therapy and blood transfusion; patients with grade IIIa complications were treated by interventions under local anesthesia; only one case with grade IIIa complications (respiratory failure) was managed through a short ICU stay, with a good outcome.

In our experience, a tracheostomy was necessary in 60% of patients, performed simultaneously in the thoracotomy/cervicotomy procedure, to ensure protection and patency of the airway, which could become obstructed and compressed due to fistulization and edema caused by severe inflammation. To this extent, it is crucial to ensure control of the airway as soon as possible and therefore the patient’s respiratory function, to prevent the onset of serious complications in the context of an already critical situation [5,9].

Moreover, in 4 patients (27%), a temporary stay in ICU occurred, since two of them were septic already upon arrival at the hospital; one patient, transferred from another hospital, presented with a complicated airway situation with pharyngo-laryngeal swelling and compression of epiglottis, and another patient (already mentioned above) developed respiratory failure after cervicotomy and thoracotomy. In fact, this group of patients is more likely to develop life-threatening complications, as reported by Mejzlik et al. [15]. 

The mean duration of chest (pleural and/or mediastinal) tubes was about 19 days; therefore, in general, drainage remained in place for a long time, as further confirmation of the severity of the disease and of the need for keeping them longer than other types of thoracotomy surgery, as they should be removed after complete resolution of all infectious collections. For this purpose, radiological post-operative monitoring by neck-chest CT scan is essential. Nevertheless, the mean duration of chest tubes in our experience was slightly longer than that reported by other authors like Yanik et al., of about 13 days [3]: this difference could be explained since in most cases these authors used less invasive approaches, which reduced overall post-operative recovery times, but on the other hand they reported a mortality rate of 15%; moreover, even if chest tube duration was slightly longer than that of other authors, as we removed drainage tubes when the amount of fluid per day was less than 200 mL and yellowish (without blood or purulent material) and this attitude might seem overly cautious, we did not experience any re-admissions to the hospital due to post-operative recurrence of pleural effusion and/or mediastinitis. 

The mean time of hospitalization was about 29 days and the outcome of surgery was successful in all patients, with no mortality rate. In this regard, it must be emphasized that hospital stay is quite long due to the severity of the disease, which requires longer recovery times, and that in our experience we had no mortality compared to the literature, reporting mortality rates of about 11–40% [5], even though our hospitalization time was slightly longer than that reported by other authors, of about 20–25 days [3,4,12,13].

Regarding the type of germs, which were isolated on the samples taken intra-operatively or post-operatively (blood culture, bronchial lavage, pleural fluid), we predominantly observed positivity for bacteria of the genus *Staphylococcus* (53%) (most frequently *S. epidermidis* and *S. aureus*) and *Streptococcus* (13%). This is partially in agreement with the literature where in most cases a higher incidence of Streptococci is described [3,4,16]. 

It is important to specify that in 20% of cases, a positivity for fungi of the *Candida* genus was found, namely *C. albicans* and *C. parapsilosis*, and in all *Candida* positive cases, there was an association with a bacterium.

In agreement with other authors [11,17], there was polymicrobial infection in about 47% of cases. This evidence is important because the co-presence of multiple micro-organisms favors the spread of the infectious process, worsening the outcome of patients. 

Finally, in 27% of our cases, no bacteria and/or fungus was detected at microbiological examination, probably due to the early establishment of proper antimicrobial therapy. 

Concerning the antimicrobial therapy, established on the basis of the antibiogram/antimicogram where available and in all cases after consultation with the infectious disease specialist, who also monitored the progress of treatment, we observed a prevalence of meropenem, metronidazole, teicoplanin, third-generation cephalosporins and clyndamicin, in a similar way to other authors [4,12,13], while the most commonly used anti-fungal drugs were fluconazole, caspofungin and anidulafungin. 

In this regard it should be emphasized that, thanks to this multidisciplinary approach in cooperation with the infectious disease specialist, a greater and faster effectiveness of antimicrobial therapy was achieved; in fact no complications related to the administration of antimicrobial treatments were described and no therapy had to be discontinued prematurely due to side effects from drugs. On the other hand, in cases of DNM, antimicrobial therapy should always be associated with the proper surgical treatment, otherwise it alone would be completely ineffective. In addition, teamwork with infectious disease specialists is also essential to increase mutual knowledge and implement good clinical practice on the rational use of antibiotics [18]. This supports measures to tackle antibiotic resistance, which is also growing during the SARS-CoV-2 pandemic [19].

Considering pre-operative risk factors or co-morbidities, we observed smoking and/or COPD in 40% of patients, overweight in 27%, diabetes mellitus in 20% and cardiopathy/vasculopathy in 20%, in agreement with the literature, although in most studies the most frequent risk factor is diabetes mellitus [1,4,9,12]. In our experience instead, the most common risk factor was smoking and/or COPD, which probably, reducing the muco-ciliary clearance and the trophism of the airway, facilitates microbial engraftment at the level of the oro-pharyngo-laryngeal mucosa and therefore infections in the typical sites of origin of the DNM. Moreover, cardiovascular diseases may favor DNM, because they cause a reduction in peripheral vascularization and thus in the trophism of muscles and fasciae, which are easily overcome by micro-organisms and therefore by the infectious process, thus facilitating the downward spread of oro-pharyngeal/cervico-fascial infections. In fact, on multivariate analysis, cardiovascular diseases were statistically significantly associated with longer chest tube duration and hospital stay, thus confirming them as a risk factor for a worse course of DNM, as reported by Celakovsky et al. [20]. Furthermore, it should be noted that the mean BMI was about 24, so in general our patients were within a normal weight category (as reported also by Kimura et al. [21]); only 4 were found to be overweight and so apparently we did not observe any correlation between overweight and DMN. 

The most relevant limitation of our study is that it was retrospective with a small number of cases (15 patients in almost 20 years) but this can be explained and justified by the rarity of the disease. Therefore, we recommend multicenter studies in order to have more data available for evaluation and results with statistical significance, especially concerning the identification of prognostic risk factors that may affect the evolution and treatment of this potentially fatal condition.

## 4. Materials and Methods

We performed a retrospective study of patients who came at our hospital and were treated for DNM at our Unit of Thoracic Surgery from 2002 to 2020. The DNM diagnosis was made on the basis of the criteria defined by Estrera and colleagues [6]. 

By retrospective consultation of medical records, we analyzed the following patients’ data: age, gender, etiopathogenesis of DNM (source of infection), degree of diffusion of DNM (evaluated with Endo’s classification [7,8] on the basis of CT scans and intra-operative findings), presence or not of septic shock at diagnosis, time between diagnosis and treatment, type of surgery (thoracotomy alone or combined with cervicotomy), duration of surgery, need for tracheostomy, intensive care unit (ICU) stay, post-operative complications, duration of chest tubes, length of hospital stay, outcome of treatment, post-operative mortality, type of germs isolated, type of antibiotics used, pre-operative risk factors or co-morbidities.

Continuous variables were reported as mean with standard deviation. Nominal variables were reported as counts and percentages. Pre-operative risk factors or co-morbidities (diabetes, overweight, smoke/COPD and cardiopathy/vasculopathy) were the variables submitted for statistical multivariate analysis, to disclose risk factors for a worse course of DNM (tracheostomy, ICU stay, post-operative complications, type II DNM, longer hospital stay and chest tube duration). Significance was defined as a *p*-value < 0.05. Statistical analyses were performed using RStudio (R v3.6.2, Dark and Stormy Night).

## 5. Conclusions

DNM is a serious condition that puts the patient’s life at risk and so requires a timely and correct treatment. In most cases, it already presents at an advanced degree of diffusion, thus a rapid diagnosis, with the support of neck and chest CT, plays a fundamental role to reduce the mortality and morbidity of this disease.

The standard surgical approach consists in a lateral thoracotomy, which guarantees an optimal exposure and cleaning of the mediastinum and pleural cavity and a lower occurrence of post-operative complications, always preceded by cervicotomy and cervical drainage and eventual treatment of diseased teeth, to treat the source of origin of DNM.

In association with surgical treatment, multidisciplinary cooperation with the infectious disease specialist is fundamental to promptly administer the proper antimicrobial therapy, on the basis of the antibiogram/antimicogram where available, and monitor the progress of treatment.

Tracheostomy to maintain a patent airway and when required, temporary ICU stay, are essential because delays could have serious consequences. Any recurrence of the disease must be recognized and treated immediately, because delays in re-operation worsen the patient’s prognosis.

Ultimately, all factors analyzed in our study should be further evaluated in future multicentric studies, to better understand the expected prognosis of the individual patient and carry out the most appropriate multidisciplinary management and treatment.

## Figures and Tables

**Figure 1 antibiotics-11-00664-f001:**
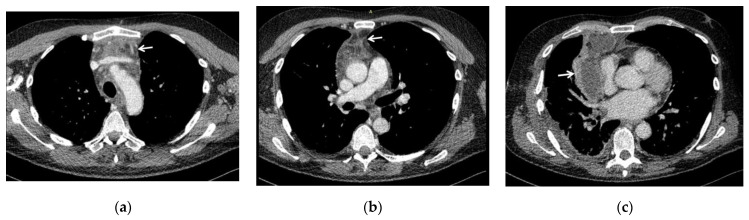
CT scans showing the spread of infection of DNM in three of our patients, evaluated according to Endo’s classification of DNM [7,8]: (**a**) type I (DNM localized in the upper mediastinum above the tracheal bifurcation); (**b**) type IIA (DNM extends to the lower anterior mediastinum, below the tracheal bifurcation); (**c**) type IIB (DNM extends to the anterior and lower posterior mediastinum), with associated right pleural effusion. White arrows indicate the mediastinal involvement.

**Figure 2 antibiotics-11-00664-f002:**
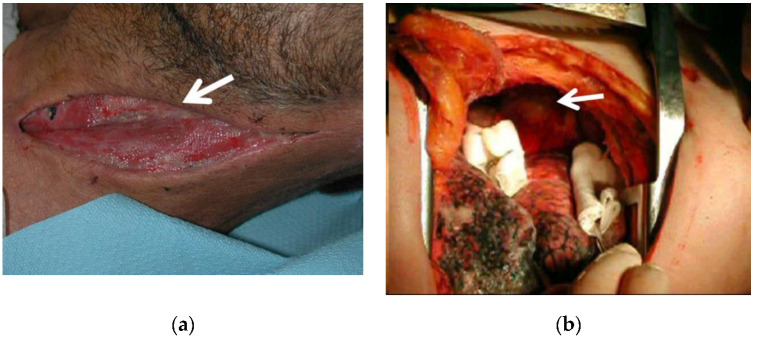
In all cases, mediastinum drainage via thoracotomy was performed after neck drainage via cervicotomy: (**a**) intra-operative field during cervicotomy showing purulent material (white arrow) coming from the cervical fasciae; (**b**) intra-operative field during thoracotomy, showing yellowish purulent collection (white arrow) in the anterior mediastinum (the lung has been retracted).

**Figure 3 antibiotics-11-00664-f003:**
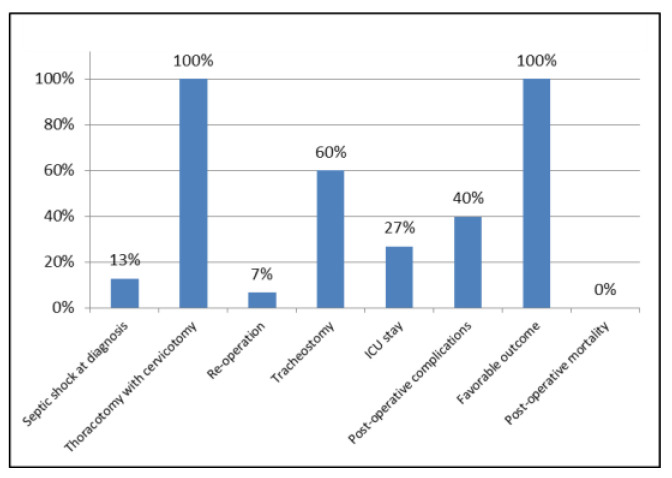
Data relating to the nominal variables, expressed as percentages.

**Table 1 antibiotics-11-00664-t001:** Data relating to the continuous variables, expressed as mean ± standard deviation (s.d.).

Continuous Variables	Mean ± S.D.
age (years)	49.07 ± 14.92
time between diagnosis and treatment (days)	2.24 ± 1.61
duration of thoracic surgery (minutes)	130.63 ± 32.34
duration of chest tubes (days)	18.91 ± 13.46
hospital stay (days)	29.33 ± 14.24
body mass index (BMI)	23.94 ± 2.67

**Table 2 antibiotics-11-00664-t002:** Data relating to the type of germs isolated on samples taken intra- or post-operatively (blood culture, bronchial lavage, pleural fluid) and antimicrobial post-operative therapy.

**Type of Germs Isolated** **on Samples Taken** **Intra-or Post-Operatively**	**53% Staphylococcus genus** (*epidermidis, aureus, hominis, capitis*)	**47%** **Polymicrobial** infection
	**13% Streptococcus genus** (*anginosus, cristatus*)	
	**27% other genus** (*K. Pneumoniae, E. coli, S. maltophilia, S. marcescens, P. aeruginosa*)	
	**20% Candida genus** (*albicans, parapsilosis*)	
**Antimicrobial Post-Operative Therapy**	**most used antibiotics**: 67% meropenem, 47% metronidazole, 40% teicoplanin, 33% third-generation cephalosporins, 27% clyndamicin	
	**anti-fungal drugs used**: fluconazole, caspofungin, anidulafungin	

**Table 3 antibiotics-11-00664-t003:** Multivariate analysis results for the risk of a worse course of DNM (*p*-value is reported, in bold when statistically significant).

	Diabetes	Overweight	Smoke/ COPD	Cardiovascular Diseases
tracheostomy	0.273	0.586	0.361	0.805
post-operative complications	0.999	0.998	0.999	0.998
type II DNM	0.551	0.094	0.058	0.664
chest tube duration > 19 days	0.843	0.765	0.562	**0.008**
hospital stay > 29 days	0.424	0.621	0.869	**0.009**

## Data Availability

The raw data are available from the corresponding author.

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
