# Peer review of "Multidisciplinary Approach in the Treatment of Descending Necrotizing Mediastinitis: Twenty-Year Single-Center Experience"

_antibiotics, 2022, doi:10.3390/antibiotics11050664_

Round 1
Reviewer 1 Report
This study performed valuable research on the rare disease descending necrotizing mediastinitis. The post-op mortality rate was zero for all fifteen patients. This shows the importance of a multidisciplinary approach to complex diseases. Looking Table 3, I was also curious about the proportion of MRSA or antibiotic-resistant bacteria. Because the antibiotic-resistant bacterial infection is more difficult to treat.
I have no further comments. Though, a few typo errors should be revised. (Ex. line 117 lengt -> length)
Reviewer 2 Report
The topic of the article is interesting a the file is sufficient. However, there are some weak points which must be addressed. Most importantly, the new information is missing, for example sone interesting statistical analysis, declaring the value of recorded factors. I would recommend introducing statistical methods and change the title of the article to more interesting. For example, “Analysis of factors of descendent......”or similar. I would recommend placing the variables results section into tables.
Then there are some grammatical and structural mistakes:
Row 30, 39 cervicotomy
....revision of parapharyngeal space is more appropriate?
Row 65, 66 This section may be divided by subheadings. It should provide a concise and precise description of the experimental results, their interpretation, as well as the experimental 66 conclusions that can be drawn...
I guess it was the matter of internal institutional corrections. I recommend omit those sentences.
Row 72 Emiglosectomy
….. means hemiglosectomy ?
Lastly, I would recommend using some recent literature to draw more complex image of this dangerous clinical unit. It might be of readers' benefit.
Mejzlik, J., Celakovsky, P., Tucek, L., Kotulek, M., Vrbacky, A., Matousek, P., . . . Chrobok, V. (2017). Univariate and multivariate models for the prediction of life-threatening complications in 586 cases of deep neck space infections: retrospective multi-institutional study. Journal of Laryngology and Otology, 131(9), 779-784. doi:10.1017/s0022215117001153
Celakovsky, P., Kalfert, D., Smatanova, K., Tucek, L., Cermakova, E., Mejzlik, J., . . . Hoskova, T. (2015). Bacteriology of deep neck infections: analysis of 634 patients. Australian dental journal, 60(2), 212-215. doi:10.1111/adj.12325
Celakovsky, P., Kalfert, D., Tucek, L., Mejzlik, J., Kotulek, M., Vrbacky, A., . . . Pasz, A. (2014). Deep neck infections: risk factors for mediastinal extension. European Archives of Oto-Rhino-Laryngology, 271(6), 1679-1683. doi:10.1007/s00405-013-2651-5
Reviewer 3 Report
Congratulations on your data.
I have several questions as the following.
- The order of subheading in the manuscript confused me. Is “Materials and Method” should be placed before the “Result”? In addition, the first paragraphy in the result, line 65-67 should be removed as it was the template!
- Regarding the post-op complications, please use the Clavien-Dindo classification to better illustrate its severity and relevant management.
- In Table 2, I suggest you design a histogram to demonstrate your result in a more straightforward manner.
- I am wondering the timing of tracheostomy, which was not mentioned in your article. Was it performed simultaneously in the thoracotomy/cervicotomy procedure? Or performed when the condition worsened in a staged procedure?
- The chest tube drainage duration was a bit tedious and prolonged. Can you provide the data of the drainage amount/characteristic per day before the removal? What’s your routine for tube removal? As the prolonged tube-indwelling days directly impacted on the length of hospital stay.
- Statistic method and software should be added in the Methods.
- I suggest you to place some operative figures to delineate the severity of DNM.
- Typos were found. For example, Line 30 (it should be” thoracotomy followed by cervicotomy"), line 39 (“thoracotomy”), and line 111(“thyroid”). Overall, the entire manuscript should be gone through carefully before the submission.
Round 2
Reviewer 2 Report
The revision made a great improvement in all aspects of the manuscript. The authors made their corrections with respect to reviver suggestions. I like the plain style of the document addressing the key messages to the readers. In this form I have no hesitation to recommend it for publication. I wish the success in author's diligent work.
Reviewer 3 Report
The authors answered comprehensively to the comments addressed.
Please make minor adjustments to the fig 1 and 2 with a neat and even format.
Congrats to you on this nice work.